# In Silico Analysis of miRNA-mRNA Binding Sites in *Arabidopsis thaliana* as a Model for Drought-Tolerant Plants

**DOI:** 10.3390/plants14121800

**Published:** 2025-06-12

**Authors:** Yryszhan Zhakypbek, Aizhan Rakhmetullina, Zhigerbek Kamarkhan, Serik Tursbekov, Qingdong Shi, Fei Xing, Anna Pyrkova, Anatoliy Ivashchenko, Bekzhan D. Kossalbayev, Ayaz M. Belkozhayev

**Affiliations:** 1Department of Mine Surveying and Geodesy, Institute Mining and Metallurgical Institute Named After O.A. Baikonurov, Satbayev University, Almaty 050043, Kazakhstan; y.zhakypbek@satbayev.university; 2Institute of Biochemistry and Biophysics, Polish Academy of Sciences, 02-106 Warsaw, Poland; arakhmet@ibb.waw.pl; 3Department of Mechanical Engineering, Institute of Energy and Mechanical Engineering Named After A. Burkitbayev, Satbayev University, Almaty 050013, Kazakhstan; zh.kamarkhan@satbayev.university; 4College of Ecology and Environment, Xinjiang University, Urumqi 830017, China; shiqd@xju.edu.cn; 5Key Laboratory of Oasis Ecology, Xinjiang University, Urumqi 830017, China; 6College of Geography and Remote Sensing Sciences, Xinjiang University, Urumqi 830017, China; xingfei@xju.edu.cn; 7Department of Biotechnology, Al-Farabi Kazakh National University, Almaty 050040, Kazakhstan; anna.pyrkova@kaznu.kz (A.P.); a.iavashchenko@gmail.com (A.I.); 8Center for Bioinformatics and Nanomedicine, Almaty 050060, Kazakhstan; 9Ecology Research Institute, Khoja Akhmet Yassawi International Kazakh Turkish University, Turkistan 161200, Kazakhstan; 10Tianjin Institute of Industrial Biotechnology, Chinese Academy of Sciences, Tianjin 300308, China; 11Department of Chemical and Biochemical Engineering, Geology and Oil-Gas Business Institute Named After K. Turyssov, Satbayev University, Almaty 050043, Kazakhstan; 12M.A. Aitkhozhin Institute of Molecular Biology and Biochemistry, Almaty 050000, Kazakhstan

**Keywords:** *Arabidopsis thaliana*, miRNA, mRNA, drought tolerance, binding sites

## Abstract

Drought stress limits plant survival and yield in arid regions. Uncovering the molecular mechanisms of drought tolerance is key to developing resilient crops. This study used *Arabidopsis thaliana* as a model to perform an in silico analysis of miRNA–mRNA interactions linked to post-transcriptional drought response. Using the MirTarget program, 274 miRNAs and 48,143 gene transcripts were analyzed to predict high-confidence miRNA–mRNA interactions based on binding free energies (−79 to −129 kJ/mole). Predicted binding sites were located in the CDS, 5′UTR, and 3′UTR regions of target mRNAs. Key regulatory interactions included ath-miR398a-c and ath-miR829-5p targeting ROS detoxification genes (*CSD1*, *FSD1*); ath-miR393a/b-5p and ath-miR167a-c-5p targeting hormonal signaling genes (*TIR1*, *ARF6*); and the miR169 family, ath-miR414, and ath-miR838 targeting drought-related transcription factors (NF-YA5, DREB1A, WRKY40). Notably, ath-miR414, ath-miR838, and the miR854 family showed broad regulatory potential, targeting thousands of genes. These findings suggest the presence of conserved regulatory modules with potential roles in abiotic stress tolerance. While no direct experimental validation was performed, the results from *Arabidopsis thaliana* provide a useful genomic framework for hypothesis generation and future functional studies in non-model plant species. This work provides a molecular foundation for improving drought and salt stress tolerance through bioinformatics-assisted breeding and genetic research.

## 1. Introduction

Drought is one of the major abiotic stress factors that significantly impact plant growth, productivity, and ecosystem stability [1,2]. Plants have evolved complex responses to withstand water deficit, including rapid stomatal closure, accumulation of osmoprotectants, activation of antioxidant enzymes, hormonal signaling changes, and long-term morphological adjustments [3]. Due to global climate change, environmental challenges are intensifying, particularly in coastal and semi-arid regions near large water bodies such as seas and lakes [4]. In these areas, shifts in precipitation patterns, increased evaporation rates, and elevated soil salinization exacerbate drought stress, posing significant threats to plant biodiversity and agricultural sustainability [5,6]. One such group of regions where these ecological changes are prominently observed includes the Caspian lowlands, the Mediterranean Basin, and Central Asian semi-deserts, where a combination of natural climatic fluctuations and anthropogenic influences has led to increased water scarcity and soil degradation [7,8]. Plant adaptation to drought is regulated through complex genetic and physiological mechanisms, many of which remain incompletely understood across different plant species [9]. Identifying the molecular basis of drought tolerance is crucial for understanding stress adaptation mechanisms and developing effective strategies to enhance plant resilience [10,11]. However, the limited genomic characterization of many drought-adapted plant species from arid and saline environments presents significant challenges in studying these stress response processes. Therefore, model organisms such as *Arabidopsis thaliana* (*A. thaliana*) serve as essential reference systems for investigating genetic pathways associated with drought response [12,13]. *A. thaliana* is a widely studied model organism in molecular biology due to its fully annotated genome, short life cycle, and extensive research history [14]. This plant possesses well-characterized drought-responsive genes, along with miRNA (microRNA, also referred to as mRNA-inhibitory RNA) regulatory networks that modulate gene expression in response to stress [15,16]. miRNAs function as key post-transcriptional regulators, influencing plant adaptation to drought by modulating the activity of essential transcription factors and signaling pathways [17,18]. A number of miRNAs are differentially expressed during drought and other abiotic stresses, implicating them in adaptive responses [19]. For example, drought upregulates or downregulates specific miRNAs that control transcription factors and enzymes in stress pathways [20]. In *A. thaliana*, several conserved miRNA-target modules are known to modulate drought tolerance. Notably, an abscisic acid (ABA) -dependent drought pathway involves the downregulation of miR159 and miR169, leading to an increased expression of their targets (Myeloblastosis (MYB) and nuclear factor Y subunit A (NF-YA) transcription factors, respectively), which promote stress tolerance [21]. Likewise, miR398 is downregulated under oxidative and drought stress, allowing for the upregulation of *CSD1*/*CSD2* (Cu/Zn superoxide dismutase) genes that detoxify reactive oxygen species (ROS) [22]. By contrast, some miRNAs are induced by drought; for instance, miR397 levels rise under drought and salinity stress, leading to a repression of laccase genes and altered lignin deposition, a response linked to improved water transport and pathogen defense [23,24]. This study aims to employ in silico approaches to analyze the miRNA-mRNA interactions involved in drought tolerance mechanisms in *A. thaliana*. The obtained results will facilitate comparative analyses of similar mechanisms in drought-adapted plant species, thereby providing insights into the conservation of stress-regulatory networks across diverse plant taxa (Figure 1). By establishing molecular connections between model plants and less-characterized plant species, this research can serve as a foundation for future genomic and biotechnological applications, including the breeding and conservation of drought-resistant plant species.

## 2. Results

### 2.1. Predicted miRNA–mRNA Interactions Targeting Key ROS Detoxification Genes Involved in Drought Stress Response in A. thaliana

Based on the MirTarget program, potential interactions between 274 miRNAs specific to *A. thaliana* and 48,143 mRNA genes were analyzed in an in silico format. The results of the analysis showed that the binding free energy (ΔG, kJ/mole) of the predicted miRNA–mRNA pairs ranged from −79 to −129 kJ/mole, indicating strong reliable interactions between the molecules. Additionally, the ΔG/ΔGm, % ratio was determined to range from 80% to 100%, reflecting a high probability of effective base pairing at the predicted target sites.

ath-miR398a-3p, ath-miR398b,c-3p, ath-miR829-5p, ath-miR426, ath-miR842, ath-miR865-3p, ath-miR396b-5p, and ath-miR824-5p were identified to interact with the mRNAs of the *CSD1*, *FSD1*, *CAT1*, *DHAR1*, *GPX1*, *PRXQ,* and *GR1* genes, which belong to the ROS detoxification pathway associated with drought tolerance in *A. thaliana*. The predicted miRNA–mRNA interactions exhibited binding energy, with ΔG/ΔGm values ranging from 80% to 87%, indicating a strong likelihood of post-transcriptional regulation under oxidative stress conditions (Table 1).

The *CSD1* gene, which encodes a Cu/Zn superoxide dismutase, was predicted to contain BSs for ath-miR398a and ath-miR398b/c in the 5′UTR at nucleotide position 118. The corresponding binding free energies were −98 kJ/mole and −93 kJ/mole, with ΔG/ΔGm ratios of 87% and 81%, respectively. These values indicate a highly stable and functionally relevant interaction, suggesting that *CSD1* is tightly regulated post-transcriptionally by conserved members of the miR398 family. The *FSD1* gene was predicted to be targeted by ath-miR829-5p, with five distinct BSs in the CDS region at nucleotide positions 623, 480, 617, 556, and 717 nt. All five sites exhibited uniform binding energies of −83 kJ/mole and ΔG/ΔGm ratios of 81%, indicating a consistent interaction strength. The presence of multiple BSs suggests potential regulatory redundancy, which may contribute to a robust post-transcriptional repression of *FSD1* during oxidative stress responses. The *CAT1* gene, encoding a catalase enzyme, was found to be targeted by ath-miR426, with a single BS located at position 1637 nt in the CDS. The binding energy of −85 kJ/mole and ΔG/ΔGm of 82% support a likely inhibitory interaction. The *DHAR1* and *GPX1* genes, both key components of the ascorbate–glutathione cycle, were predicted to be targeted in CDS by ath-miR842 and ath-miR865-3p, respectively. The predicted binding energies were −91 kJ/mole for *DHAR1* and −83 kJ/mole for *GPX1*, with corresponding ΔG/ΔGm ratios of 81% and 83%, indicating an effective and potentially functional post-transcriptional regulation of these antioxidant enzymes under oxidative stress conditions. Two isoforms of the *PRXQ* gene were predicted to be targeted by ath-miR396b-5p at two CDS positions (455 nt and 193 nt), with binding energies of −87 kJ/mole and ΔG/ΔGm ratios of 82%. *GR1*, a glutathione reductase gene, showed a predicted interaction with ath-miR824-5p in the 3′UTR region at position 2083 nt, with a ΔG value of −87 kJ/mole and a ΔG/ΔGm ratio of 80%, indicating moderate but potentially functional regulation.

In this study, we observed both canonical and noncanonical base pairings in the predicted miRNA–mRNA interactions, illustrating binding between selected antioxidant-related genes and their corresponding miRNAs involved in the drought-responsive ROS detoxification pathway in *A. thaliana* (Figure 2). Canonical A–U and G–C pairings dominated most alignments, contributing to high binding affinity (ΔG ranging from −83 to −98 kJ/mole). Nonetheless, noncanonical matches such as G–U and A–C were also identified, particularly in miRNAs such as ath-miR829-5p, ath-miR426, ath-miR842, ath-miR865-3p, and ath-miR396b-5p. Despite the presence of these mismatches, the overall ΔG/ΔGm ratios remained high (80–87%), suggesting that these noncanonical sites still support biologically meaningful interactions.

Among the miRNAs targeting key antioxidant genes involved in ROS detoxification, miR398 is already well-established as a critical post-transcriptional regulator of redox signaling in plants, primarily through its interaction with *CSD1* and *CSD2* genes [21]. Our in silico findings not only confirmed this regulatory relationship but also revealed an expanded network of stress-related gene targets. For instance, ath-miR398a-3p exhibited predicted BSs across 41 genes, while ath-miR398b,c-3p targeted 13 genes, with high binding energy ranging from −91 to −115 kJ/mole. Notably, several of the genes predicted to interact with ath-miR398a-c-3p are functionally linked to drought stress adaptation in *A. thaliana*. These include *NHX4*, which plays a key role in salt stress tolerance through ion homeostasis [25]; *SOD1*, which encodes a superoxide dismutase critical for ROS scavenging under oxidative stress [26]; and *THE1*, a receptor-like kinase involved in cell wall integrity sensing and growth regulation [27,28].

Overall, the identified ath-miR398a-3p, ath-miR398b,c-3p, ath-miR829-5p, ath-miR426, ath-miR842, ath-miR865-3p, ath-miR396b-5p, and ath-miR824-5p were found to interact with genes involved in key antioxidant and drought-response pathways. These findings are not only valuable for elucidating drought tolerance mechanisms in *A. thaliana*, the chosen model organism, but also provide a foundation for comparative genomic and regulatory studies in drought-adapted plant species from arid and saline environments. Species which have naturally evolved in arid climates remain largely underexplored at the genomic and post-transcriptional levels. Therefore, the miRNA–mRNA interaction networks identified in *A. thaliana* may serve as a reference model, offering insights into conserved regulatory elements and guiding future in silico and experimental studies aimed at understanding and enhancing drought resilience in these ecologically significant plant species.

### 2.2. Characteristics of miRNA Binding Sites in mRNAs of Drought-Responsive Hormonal Signaling Genes in A. thaliana

During drought stress, plant hormonal signaling pathways undergo extensive reprogramming, primarily involving ABA, as well as auxin, gibberellins, ethylene, brassinosteroids, and other phytohormones. miRNAs act as key post-transcriptional regulators in these pathways, linking hormonal signals to the expression of specific target genes. Our analysis identified BSs for 19 miRNAs within the mRNA sequences of key genes involved in hormonal signaling pathways, including *ABI2*, *ARF2*, *ARF6*, *CKX1*, *EIN2*, *EIN3*, *HK2*, *JAZ1*, *PYL4*, *RCAR1*, *RCAR3*, *RD22*, and *TIR1*. The predicted miRNA-mRNA interactions exhibited free binding energies ranging from −81 to −106 kJ/mole, with ΔG/ΔGm ratios between 80% and 91%, indicating high thermodynamic stability and a strong likelihood of biological relevance (Table 2). These interactions provide important insights into the post-transcriptional regulation of hormone-mediated responses under drought stress at the molecular level.

Based on the predicted miRNA BSs in the mRNAs of hormonal signaling pathway genes involved in the drought response in *A. thaliana*, several miRNAs were found to bind within the same region of a single gene. For instance, the *ARF6* gene, a key regulator in auxin signaling known to modulate plant growth and stress responses [29], was predicted to interact with three distinct miRNAs—ath-miR167c-5p and ath-miR167a,b-5p. All three BSs are located in the CDS region, specifically at 3340 and 3341 nt (ΔG ranging from −91 to −98 kJ/mole). These interactions suggest potential cooperative or competitive regulation of *ARF6* by the miR167 families under drought stress.

Several complementary BSs for miRNAs were identified within the mRNA sequence of the *EIN2* gene, which plays a crucial role in regulating plant stress responses, leaf senescence, fruit ripening, and defense against pathogens [30]. Specifically, a BS for ath-miR847 was located at position 3477 within the CDS region, with a predicted binding free energy of −89 kJ/mole and a ΔG/ΔGm ratio of 82%. Additionally, four closely related miRNAs—ath-miR854a, ath-miR854b, ath-miR854c, and ath-miR854d—were each predicted to bind at position 26 in the 5′UTR of *EIN2* (NM_120406.5), with identical predicted binding energies of −93 kJ/mole and ΔG/ΔGm ratios of 81%.

The conserved ath-miR393a-5p and ath-miR393b-5p, which exhibited the highest predicted binding free energies among the analyzed miRNAs, were predicted to interact in the CDS region of *TIR1* mRNA at nucleotide position 1965. *TIR1* encodes a key receptor in the auxin signaling pathway and plays a central role in regulating gene expression in response to environmental stimuli, including drought stress.

The ath-miR854 family, comprising ath-miR854a, ath-miR854b, ath-miR854c, and ath-miR854d, was found to possess complementary BSs in the 5′UTR of *TIR1* mRNA, specifically at positions 218 and 224 nt. These interactions exhibited binding free energies ranging from −93 to −100 kJ/mole and ΔG/ΔGm ratios between 81% and 87%. The presence of multiple conserved miRNAs targeting the same regulatory region suggests a coordinated post-transcriptional control mechanism governing *TIR1* expression (Figure 3). These results collectively indicate that *TIR1* is subject to multilayered miRNA-mediated regulation, which may be essential for fine-tuning auxin sensitivity and downstream physiological responses under drought conditions in *A. thaliana*. Such interactions emphasize the broader role of miRNA regulatory networks in modulating hormone signaling pathways to support stress adaptation.

The insights gained from this *A. thaliana*-based analysis offer a valuable framework for future comparative studies, potentially guiding the identification of homologous miRNA-regulated pathways in drought-adapted plant species from arid and saline regions. Ultimately, these findings could facilitate the development of genetically informed strategies for enhancing drought tolerance in native plant populations and region-specific crop species.

### 2.3. Predicted miRNA Target Interactions Regulating Drought-Responsive Transcription Factor Genes in A. thaliana

The in silico analysis performed using the MirTarget program revealed a regulatory network comprising 33 miRNAs predicted to interact with key transcription factor genes involved in the drought stress response, including members of the AP2/ERF, NAC, bZIP, MYB, WRKY, HD-ZIP, C2H2, CAMTA, and NF-YA families (Table 3).

Among the transcription factors identified, *DREB1A*, *DREB2A*, and *ERF7*, members of the AP2/ERF family involved in the drought stress response, were targeted by ath-miR414, ath-miR838, and ath-miR418, respectively. The predicted binding free energy (ΔG) ranged from −87 to −89 kJ/mole, with relative binding efficiencies (ΔG/ΔGm) between 82% and 84%. The BSs were located in the CDS for *DREB1A* and in the 5′UTR for *DREB2A* and *ERF7*.

The *ANAC055*, *NAC083*, *NAC096*, *NAC100*, and *NAC102* genes, members of the NAC transcription factor family associated with drought tolerance mechanisms in *A. thaliana*, are targeted by ath-miR393b-3p, ath-miR1886.3, ath-miR414, ath-miR863-3p, ath-miR164a, miR164b-5p, and ath-miR158a-5p, respectively. The predicted binding free energy (ΔG) ranged from −83 to −106 kJ/mole, with relative binding efficiencies (ΔG/ΔGm) between 82% and 91% observed in both CDS and 5′UTR.

The bZIP (AREB/ABF) transcription factor family exhibited multiple miRNA interactions, with *ABF1*, *ABF3*, *bZIP60*, and *bZIP68* identified as targets of ath-miR835-3p, ath-miR169f-3p, ath-miR866-3p, ath-miR414, and ath-miR854a-d. The ΔG values for these interactions ranged from −83 to −96 kJ/mole, with ΔG/ΔGm values between 80% and 88%. Binding was detected in both CDS and 5′UTR regions, depending on the specific gene–miRNA pairing.

In the MYB transcription factor family, several miRNA–mRNA interactions associated with drought tolerance were detected. *MYB15*, *MYB60*, *MYB96*, *MYB102*, *MYB108*, and *MYB116* were identified as targets of ath-miR828, ath-miR414, ath-miR829-5p, ath-miR172b,e-5p, ath-miR834, and ath-miR858a, respectively. The predicted ΔG ranged from −85 to −96 kJ/mole, with ΔG/ΔGm values between 80% and 88%. Most BSs were located in CDS regions. These transcription factors have been implicated in drought stress responses; notably, *MYB102* has previously been reported to enhance drought tolerance through modulation of ABA signaling pathways [31], and its interaction with ath-miR829-5p (ΔG = −85 kJ/mole; ΔG/ΔGm = 83%) suggests potential post-transcriptional regulation under drought stress.

The WRKY transcription factor family displayed multiple miRNA–mRNA interactions relevant to drought stress adaptation. *WRKY18*, *WRKY25*, *WRKY33*, *WRKY40*, WRKY57, *WRKY63*, and *WRKY75* were targeted by ath-miR781a, ath-miR403-5p, ath-miR845a, ath-miR838, ath-miR156g, ath-miR868-5p, ath-miR472-5p, ath-miR855, ath-miR847, and ath-miR407. The predicted ΔG ranged from −79 to −89 kJ/mole, while ΔG/ΔGm values spanned 80% to 84%. BSs were located in CDS regions, with *WRKY18* uniquely exhibiting miRNA targeting in the 3′UTR. Among these, *WRKY40,* a transcription factor previously implicated in ABA-mediated drought signaling, demonstrated [32] interaction with ath-miR838 (ΔG = −87 kJ/mole; ΔG/ΔGm = 84%), suggesting a potential post-transcriptional regulatory role in stress adaptation.

Additional miRNA–mRNA interactions were identified across the HD-ZIP, C2H2, and CAMTA transcription factor families. The *ATHB7* and *ATHB54* genes were targeted by ath-miR865-3p and ath-miR447a.2-3p, respectively, with binding in the CDS and 5′UTR regions. *ZAT10* was targeted by ath-miR778, exhibiting interaction in CDS (ΔG = −91 kJ/mole; ΔG/ΔGm = 83%). For the CAMTA family, *CAMTA3* and *CAMTA6* were targeted by ath-miR844-3p, ath-miR2112-3p, and ath-miR857, with all BSs located in CDS regions. The predicted ΔG ranged from −83 to −91 kJ/mole, with ΔG/ΔGm values spanning 80% to 83%.

Extensive miRNA–mRNA interactions were detected within the NF-YA transcription factor family, all mediated by the miR169 family. The NF-YA genes *NF-YA1*, *NF-YA2*, *NF-YA3*, *NF-YA5*, *NF-YA6*, *NF-YA8*, *NF-YA9*, and *NF-YA10* exhibited target sites located in the 3′UTR. The predicted ΔG values ranged from −93 to −102 kJ/mole, while ΔG/ΔGm values were between 80% and 89%. Notably, multiple BSs were identified in *NF-YA1*, *NF-YA2*, and *NF-YA10*, reflecting strong regulatory potential by miR169 family members under drought stress conditions.

Among the predicted interactions, several miRNA–mRNA pairs exhibited particularly strong binding energy, with ΔG values equal to or lower than −100 kJ/mole. Importantly, these included interactions of ath-miR164a and ath-miR164b-5p targeting *NAC100* (ΔG = −106 kJ/mole, ΔG/ΔGm = 91%), all binding at position 970 in the CDS region. Multiple interactions were also identified within the NF-YA family, such as miR169 family members targeting *NF-YA1*, *NF-YA2*, *NF-YA5*, *NF-YA6*, and *NF-YA10* (ΔG = −100 to −102 kJ/mole). These strong binding interactions may represent key regulatory points in the drought stress response pathway (Figure 4A). Several miRNA molecules were found to exhibit complementary binding potential with dozens, hundreds, or even thousands of different genes. This highlights their broad regulatory capacity and suggests that they may act as key modulators capable of orchestrating complex gene expression networks at the post-transcriptional level. Among them, ath-miR414 demonstrated the highest targeting potential, with predicted interactions involving 4072 unique genes and 10,011 BSs, followed by ath-miR838 (1067 target genes, 2054 BSs) and the miR854 family (ath-miR854a-d; each with 793 unique target genes and 1146 BSs), followed by ath-miR838 (1067 unique target genes and 1095 BSs). These BSs were distributed across various regions of CDS, 3′UTR, and 5′UTR, with predicted binding free energies ranging from −85 to −106 kJ/mole. Specifically, ath-miR414 exhibited strong binding free energy (−104 to −106 kJ/mole) within the CDS region of several genes, including *AT4G04630*, *AT4G33060*, and *AT5G04980*. For *AT5G04980*, four distinct transcript variants were identified as targets: NM_120580.3, NM_001085060.2, NM_001342799.1, and NM_001342800.1 (Figure 4B). The interaction between miRNA and mRNA molecules is characterized by near-complete nucleotide complementarity and high binding free energy. The structure of these interactions involves not only canonical base pairs such as A–U and G–C but also noncanonical matches including A–C and G–U. Due to the MirTarget program’s ability to account for such noncanonical pairings, the resulting miRNA–mRNA duplex preserves a helical configuration, and stacking interactions between all bases enhance the thermodynamic stability of the complex.

A regulatory network was predicted using the MirTarget program. Heatmap visualization (Figure 5) confirmed that miR169 family members exhibited extensive regulatory interactions with key drought-responsive transcription factors in *A. thaliana*. Specifically, miR169 molecules were found to establish stable binding with major NF-YA transcription factors, all of which were among the top 10 identified regulatory hubs. The predicted binding free energies for these interactions ranged from −96 to −100 kJ/mole, with ΔG/ΔGm ratios consistently between 80% and 87%. The strongest binding interactions (ΔG = −100 kJ/mole) were observed for ath-miR169a-c-5p, each demonstrating complementary BSs within the 3′UTR of the *NF-YA5* gene. Importantly, the ability of miR169 family members to simultaneously regulate multiple transcription factors highlights a sophisticated and coordinated post-transcriptional regulatory network underlying drought stress adaptation. Collectively, these results validate the biological relevance of the in silico predictions and support the hypothesis that conserved miRNA–mRNA regulatory modules may play a critical role in enhancing drought resilience in *A. thaliana* and related plant species.

## 3. Discussion

The present study provides an extensive in silico analysis of miRNA–mRNA interactions in *A. thaliana*, a widely used model organism in plant molecular biology, with a focus on drought-responsive regulatory mechanisms relevant to arid-adapted species. Using the MirTarget program, we identified miRNA regulatory networks targeting key drought-responsive genes across multiple functional categories, including ROS detoxification, hormonal signaling, and transcription factors. While *A. thaliana* serves as a powerful model, the direct extrapolation of miRNA–mRNA interactions to drought-adapted species should be approached cautiously due to potential evolutionary divergence among plant lineages.

In the drought-responsive ROS detoxification pathway, eight miRNAs were predicted to interact with seven key antioxidant genes, including *CSD1*, *FSD1*, *CAT1*, *DHAR1*, *GPX1*, *PRXQ*, and *GR1*. The detection of conserved BSs for the miR398 family in *CSD1* corroborates previous findings linking miR398 downregulation to enhanced superoxide dismutase expression under oxidative stress [22]. These findings expand the regulatory module by identifying additional miRNAs, such as ath-miR829-5p and ath-miR842, targeting other antioxidant genes, suggesting broader post-transcriptional control of the ROS scavenging system. Our in silico analysis in *A. thaliana* predicted that miRNAs such as ath-miR398a/b/c-3p, ath-miR829-5p, and others target antioxidant and stress-related genes (*CSD1*, *FSD1*, *CAT1*, *DHAR1*, *GPX1*, *PRXQ*, and *GR1*). This aligns with the characterization of miR398 as a highly conserved miRNA widespread in angiosperms, underscoring the universality of the miR398–*CSD1* regulatory module in stress responses [33]. Given the conserved miR398–*CSD1* regulatory module observed in *A. thaliana*, and the known antioxidant responses in various halophytic species, it is hypothesized that similar miRNA-mediated regulation may operate in salt- and drought-tolerant plants. Although direct experimental data for wild halophytes are currently limited, such comparative insights offer a conceptual model for future validation and for guiding breeding strategies in stress-resilient crops.

Drought-responsive hormonal signaling genes exhibited a diverse array of miRNA–mRNA interactions, with 19 miRNAs predicted to target 13 key genes. The interaction between ath-miR393a/b-5p and *TIR1*, with a predicted binding free energy as low as −106 kJ/mole, aligns with earlier findings that miR393 modulates auxin signaling under drought stress by downregulating auxin receptor genes [34]. This regulatory mechanism is known to suppress lateral root growth, allowing plants to conserve resources during water deficit. Such hormone-associated miRNA modules appear conserved; for instance, overexpressing miR159 (which targets MYB transcription factors) enhances drought tolerance in both woody and herbaceous plants [1]. Elucidating these conserved miRNA–mRNA interactions could provide insights into the molecular basis of drought tolerance in halophytes and guide future research aimed at transferring such regulatory traits into stress-resilient crops adapted to arid and saline environments.

The characterization of 33 miRNAs predicted that they regulate 58 transcription factor genes across major families such as AP2/ERF, NAC, bZIP, MYB, WRKY, and NF-YA, which are involved in regulatory networks associated with plant tolerance to saline and arid conditions. Importantly, strong binding interactions (ΔG ≤ −100 kJ/mole) were identified for *NAC100* (ath-miR164a,b) and multiple NF-YA genes targeted by the miR169 family. This result reinforces the established role of miR169 in modulating NF-YA expression to regulate drought tolerance [35] and highlights *NAC100* as an additional high-affinity target within the drought-responsive transcriptional network. Indeed, the miR169–NF-YA module is pervasive in plants; miR169 is one of the largest miRNA families, with its targets mainly encoding NF-YA transcription factors. Experimental validation in a halophyte confirmed that miR169b targets *NFYA1* (via RLM-RACE) and that modulating this module enhances drought tolerance [36]. The detection of ath-miR414, ath-miR838, and the miR854 family as miRNAs with the broadest regulatory reach targeting thousands of genes suggests their function as potential master regulators orchestrating complex stress response pathways.

The biological effect of miRNA–mRNA interactions is not only defined by the strength of base pair complementarity but also by the specific localization of binding sites within the mRNA structure. Binding to the 5′UTR region can interfere with translation initiation, while interactions in the CDS region may inhibit translation elongation. Meanwhile, miRNA binding to the 3′UTR often impacts mRNA stability and translational efficiency [37,38]. In this study, we observed that several miRNAs target different regions of mRNA transcripts, such as the 5′UTR (e.g., ath-miR854a–d binding to *EIN2*) and CDS (e.g., ath-miR847), indicating functional diversity depending on binding site location. While this study is based on computational predictions, future analyses in non-model, stress-adapted plants may uncover conserved mechanisms with potential for crop improvement. Experimental validation such as degradome sequencing (RLM-RACE), qRT-PCR, Northern blotting, and CRISPR/Cas9 remains essential to confirm these regulatory roles [39,40,41].

This study investigates the miRNA-mediated regulatory network associated with drought tolerance in *A. thaliana*, identifying conserved post-transcriptional modules that may also function in other stress-adapted plant species. Together with targeted validation studies, these results offer actionable insights for engineering stress resilience. While further validation is necessary, these findings provide a potential framework for exploring shared molecular mechanisms of stress resilience in non-model plants.

## 4. Materials and Methods

The genomic dataset used in this study consisted of 48,143 genes from *A. thaliana*. The mRNA sequences were retrieved from the NCBI RefSeq database (RefSeq: GCF_000001735.4, June 2018) to ensure consistency and genome annotation accuracy. The corresponding nucleotide sequences of 274 miRNAs were obtained from miRBase v.22 (http://www.mirbase.org/, accessed on 10 March 2025), a comprehensive database of experimentally validated plant miRNAs.

The identification of miRNA target genes was conducted using the MirTarget program [42,43], which evaluates miRNA-mRNA interactions based on various binding parameters. This software calculates the binding free energy (ΔG, kJ/mole), the relative binding energy (ΔG/ΔGm, %), as well as the position and structural patterns of potential binding sites (BSs). The maximum binding free energy (ΔGm) is defined as the free energy associated with miRNA binding to a fully complementary nucleotide sequence. The relative free energy (ΔG/ΔGm) serves as a comparative metric to assess the strength of miRNA-mRNA interactions [42].

A distinctive feature of the MirTarget program is its ability to account for non-canonical nucleotide interactions in miRNA-mRNA binding. In addition to traditional base pairing between adenine (A) and uracil (U) and guanine (G) and cytosine (C), the algorithm also considers A-C and G-U interactions that occur via a single hydrogen bond. The program assumes that the distance between A-C and G-U pairs is equivalent to that of conventional G-C and A-U base pairs, enabling a more comprehensive assessment of miRNA binding potential [44,45,46]. When multiple miRNAs bind to the same mRNA or when the BSs of two distinct miRNAs partially overlap, the preferred BS is determined based on the higher free binding energy (ΔG), as stronger interactions indicate greater binding stability. The reliability and accuracy of the MirTarget program in identifying BSs have been validated in multiple studies [47,48,49,50], confirming its effectiveness in miRNA target prediction. Additionally, the MirTarget algorithm demonstrates equivalent efficiency in predicting BSs for plant miRNAs, ensuring accurate identification of miRNA-mRNA interactions across various plant species.

Following miRNA–mRNA target prediction, data visualization and clustering were performed using TBtools v1.120 (https://github.com/CJ-Chen/TBtools, accessed on 10 March 2025) [51]. Heatmaps based on binding free energy values (ΔG) were generated to highlight key drought-responsive regulatory hubs, with color gradients indicating interaction strength.

## 5. Conclusions

This study provides a comprehensive in silico analysis of miRNA–mRNA interactions regulating drought-responsive genes in *A. thaliana*, revealing key post-transcriptional regulatory networks across antioxidant defense, hormonal signaling, and transcription factor pathways. Our findings identified 8 miRNAs targeting 7 ROS detoxification genes, 19 miRNAs regulating 13 hormonal signaling genes, and 33 miRNAs targeting transcription factor genes across families including AP2/ERF, NAC, bZIP, MYB, WRKY, HD-ZIP, CAMTA, and NF-YA, all of which are key components of drought-responsive regulatory pathways. These results suggest the potential conservation of miRNA-mediated regulatory mechanisms across drought- and salt-tolerant plant species inhabiting arid and saline environments. By leveraging *A. thaliana* as a genomic reference, this work generates testable hypotheses for non-model species and offers molecular insights for improving drought resilience in plants. However, as this research is based solely on in silico predictions, experimental validation through qRT-PCR, degradome sequencing, and functional assays is essential to confirm the biological roles of the predicted interactions. This study lays the groundwork for integrating bioinformatics and experimental approaches toward enhancing stress tolerance in crops and conserving keystone species in arid ecosystems.

## Figures and Tables

**Figure 1 plants-14-01800-f001:**
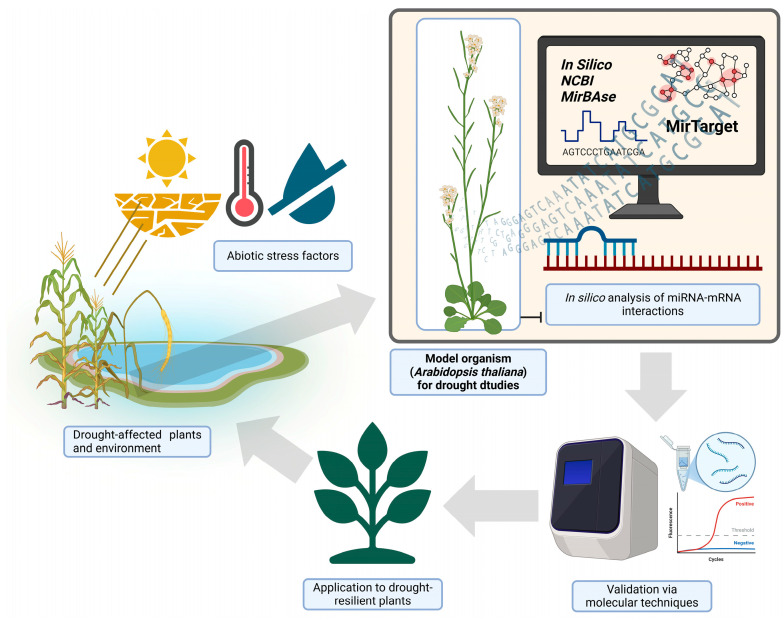
Experimental workflow for identifying drought tolerance mechanisms in *A. thaliana* as a model organism. Abiotic stress factors: depiction of environmental stressors such as high temperature, drought, and soil dryness. Drought-affected plants and environment: illustration of plants experiencing drought stress in a natural setting. Model organism (*A. thaliana*): shown as the central focus of the study for understanding drought tolerance. In silico analysis: computational approaches, including databases like NCBI, MirBase, and MirTarget, for predicting miRNA-mRNA interactions. Validation via molecular techniques: depiction of molecular tools (e.g., qRT-PCR) used to confirm in silico predictions. Application to drought-resilient plants: translation of research findings to improve drought tolerance in other plant species. Created with BioRender (https://biorender.com, accessed on 10 March 2025), License No. VW280GN3IN.

**Figure 2 plants-14-01800-f002:**
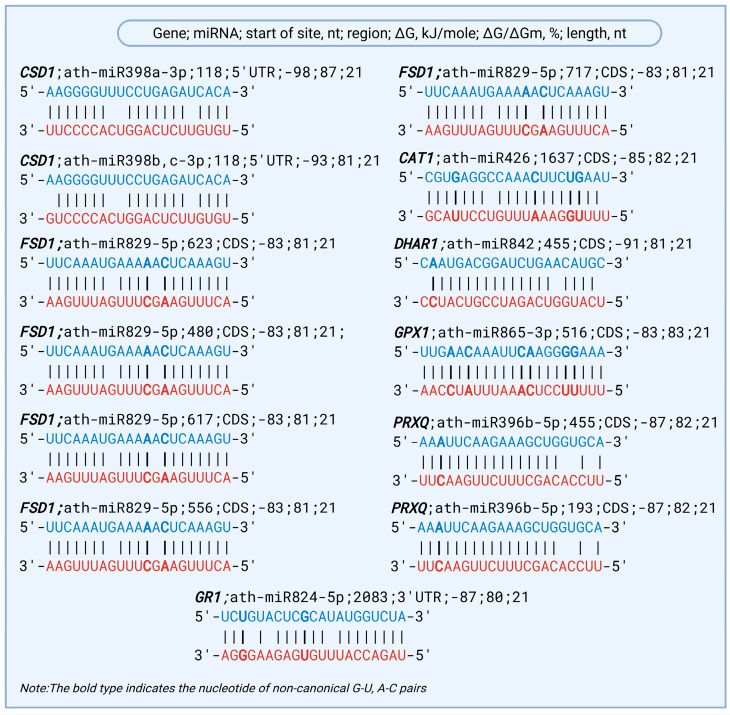
The schemes of predicted base-pairing interactions between miRNAs and antioxidant genes involved in drought-responsive ROS detoxification. Note: Blue nucleotides indicate mRNA sequences, while red nucleotides represent miRNAs.

**Figure 3 plants-14-01800-f003:**
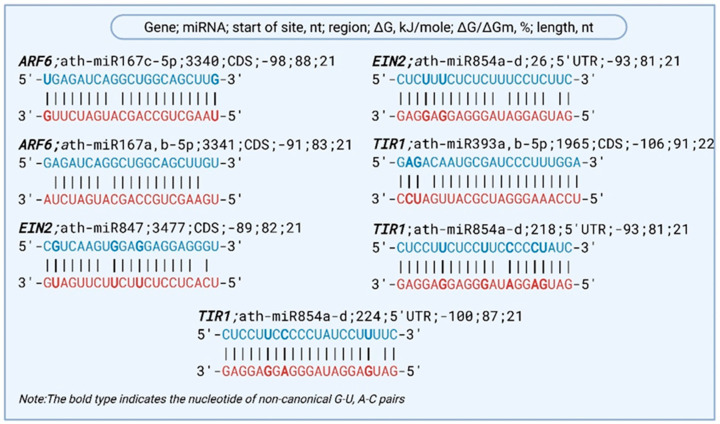
Predicted miRNA–mRNA interactions regulating hormonal signaling genes involved in drought response. Note: Blue nucleotides indicate mRNA sequences, while red nucleotides represent miRNAs.

**Figure 4 plants-14-01800-f004:**
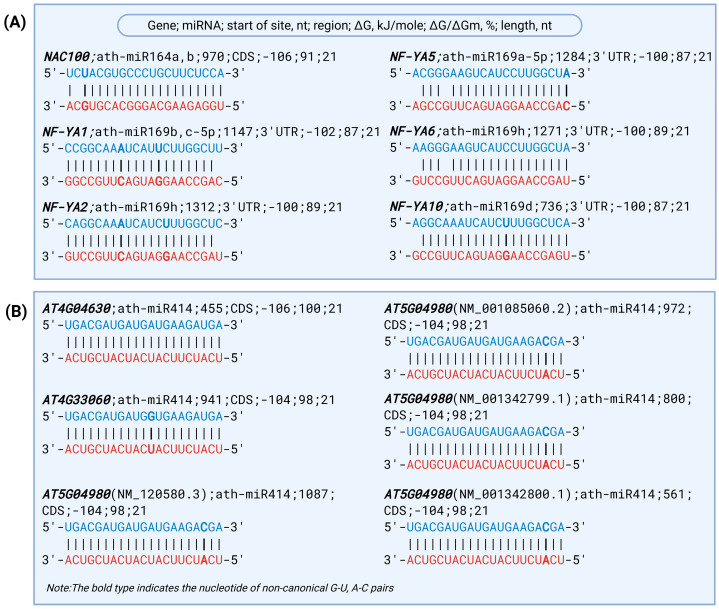
The schemes of predicted base-pairing interactions between miRNAs and their target mRNAs. (**A**) Predicted interactions between miRNAs (ath-miR164a,b, ath-miR169b,c-5p, ath-miR169h, ath-miR169a-5p, ath-miR169d) and the target mRNAs of NF-YA transcription factors; (**B**) predicted ath-miR414 BSs of *AT4G04630*, *AT4G33060*, and *AT5G04980* genes. Note: Blue nucleotides indicate mRNA sequences, while red nucleotides represent miRNAs.

**Figure 5 plants-14-01800-f005:**
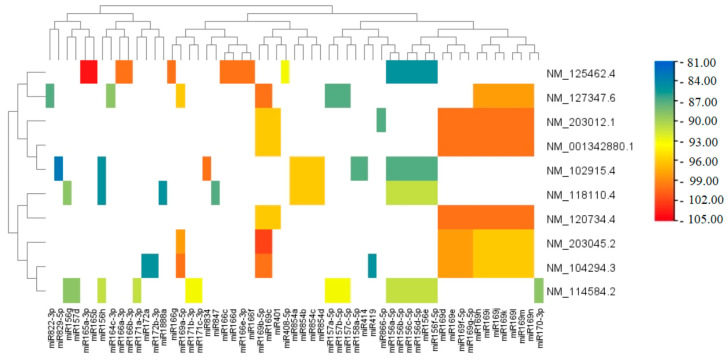
Top 10 genes targeted by multiple miRNAs: heatmap analysis of binding free energies. The heatmap visualizes predicted miRNA–mRNA interactions based on free energy values (ΔG, kJ/mole). Red colors represent stronger binding, yellow and green colors represent moderate binding, and white areas indicate no predicted binding between the miRNA and mRNA.

**Table 1 plants-14-01800-t001:** Predicted characteristics of miRNA binding sites in mRNAs of ROS detoxification pathway genes involved in drought tolerance.

Gene	NCBI Reference Sequence	miRNA	Start of Site, nt	Region	ΔG, kJ/mole	ΔG/ΔGm, %	Protein/Function
*CSD1*	NM_100757.4	ath-miR398a-3p	118	5′UTR	−98	87	Cu/Zn superoxide dismutase—detoxifies superoxide radicals
	NM_100757.4	ath-miR398b,c-3p	118	5′UTR	−93	81	
*FSD1*	NM_179109.3	ath-miR829-5p	623	CDS	−83	81	Fe superoxide dismutase—detoxifies superoxide radicals
	NM_179110.2	ath-miR829-5p	480	CDS	−83	81	
	NM_118642.2	ath-miR829-5p	617	CDS	−83	81	
	NM_001203905.1	ath-miR829-5p	556	CDS	−83	81	
	NM_001036633.2	ath-miR829-5p	717	CDS	−83	81	
*CAT1*	NM_101914.4	ath-miR426	1637	CDS	−85	82	Catalase 1—breaks down hydrogen peroxide
*DHAR1*	NM_101814.5	ath-miR842	455	CDS	−91	81	Dehydroascorbate reductase 1—regenerates ascorbate
*GPX1*	NM_128065.5	ath-miR865-3p	516	CDS	−83	83	Glutathione peroxidase 1—reduces hydrogen peroxide and lipid peroxides
*PRXQ*	NM_001203050.1	ath-miR396b-5p	455	CDS	−87	82	Peroxiredoxin Q—reduces peroxides in chloroplasts
	NM_001338777.1	ath-miR396b-5p	193	CDS	−87	82	
*GR1*	NM_113322.5	ath-miR824-5p	2083	3′UTR	−87	80	Glutathione reductase 1—regenerates GSH

**Table 2 plants-14-01800-t002:** Predicted characteristics of miRNA BSs in mRNAs of hormonal signaling pathway genes involved in drought responses.

Gene	NCBI Reference Sequence	miRNA	Start of Site, nt	Region	ΔG, kJ/mole	ΔG/ΔGm, %	Protein/Function
*ABI2*	NM_001125976.2	ath-miR781a	164	5′UTR	−87	85	Protein phosphatase 2C—negative regulator of ABA signaling
*ARF2*	NM_001203662.1	ath-miR866-3p	308	5′UTR	−85	85	Auxin response factor 2—represses auxin-responsive gene expression
*ARF6*	NM_102771.4	ath-miR167c-5p	3340	CDS	−98	88	Auxin response factor 6—regulates auxin-responsive gene expression
	NM_102771.4	ath-miR167a,b-5p	3341	CDS	−91	83	
*CKX1*	NM_001336920.1	ath-miR407	939	CDS	−81	83	Cytokinin oxidase/dehydrogenase 1—degrades cytokinins
	NM_001336920.1	ath-miR870-3p	2050	3′UTR	−85	82	
	NM_001336920.1	ath-miR390b-3p	2070	3′UTR	−87	80	
*EIN2*	NM_120406.5	ath-miR847	3477	CDS	−89	82	Ethylene-insensitive 2—central regulator of ethylene signaling
	NM_120406.5	ath-miR854a-d	26	5′UTR	−93	81	
*EIN3*	NM_112968.4	ath-miR172e-3p	1589	CDS	−89	82	Ethylene-insensitive 3—activates ethylene-responsive transcription
	NM_112968.4	ath-miR2111b-3p	759	CDS	−89	81	
*HK2*	NM_122966.3	ath-miR835-5p	1075	CDS	−87	85	Histidine kinase 2—cytokinin receptor in signaling
	NM_122966.3	ath-miR858a	3525	CDS	−87	80	
*JAZ1*	NM_001332386.1	ath-miR847	355	5′UTR	−87	80	Jasmonate ZIM-domain protein 1—represses jasmonate-responsive transcription
*PYL4*	NM_129387.3	ath-miR1886.3	879	CDS	−81	83	ABA receptor—mediates abscisic acid stress responses
*RCAR1*	NM_100018.5	ath-miR840-3p	126	5′UTR	−85	80	ABA receptor—inhibits PP2Cs in response to abscisic acid
*RCAR3*	NM_124695.4	ath-miR866-3p	180	5′UTR	−81	81	ABA receptor—inhibits PP2Cs upon ABA perception
	NM_124695.4	ath-miR773a	416	5′UTR	−87	80	
*RD22*	NM_122472.4	ath-miR398a-5p	109	5′UTR	−89	81	ABA-inducible protein—involved in drought and dehydration response
*TIR1*	NM_116163.4	ath-miR393a,b-5p	1965	CDS	−106	91	Auxin receptor—mediates degradation of Aux/IAA repressors
	NM_116163.4	ath-miR854a-d	218	5′UTR	−93	81	
	NM_116163.4	ath-miR854a-d	224	5′UTR	−100	87	

**Table 3 plants-14-01800-t003:** Predicted characteristics of miRNA target interactions regulating drought-responsive transcription factor genes.

Transcription Factors	Gene	NCBI Reference Sequence	miRNA	Start of Site, nt	Region	ΔG, kJ/ Mole	ΔG/ΔGm, %
AP2/ERF Family	*DREB1A*	NM_118680.2	ath-miR414	1009	CDS	−89	84
	*DREB2A*	NM_001036760.1	ath-miR838	62	5′UTR	−87	84
	*ERF7*	NM_112922.3	ath-miR418	154	5′UTR	−87	82
NAC Family	*ANAC055*	NM_112418.4	ath-miR393b-3p	7	5′UTR	−89	84
	*NAC083*	NM_121321.4	ath-miR1886.3	1137	CDS	−83	85
		NM_121321.4	ath-miR414	888	CDS	−89	84
	*NAC096*	NM_124029.3	ath-miR863-3p	874	CDS	−85	82
	*NAC100*	NM_001345474.1	ath-miR164a,b	970	CDS	−106	91
	*NAC102*	NM_001345612.1	ath-miR158a-5p	389	CDS	−87	85
bZIP Family	*ABF1*	NM_001198254.2	ath-miR835-3p	109	5′UTR	−87	80
	*ABF3*	NM_001036708.3	ath-miR169f-3p	450	CDS	−96	83
		NM_001036708.3	ath-miR866-3p	677	CDS	−83	83
	*bZIP60*	NM_103458.3	ath-miR414	388	CDS	−93	88
		NM_103458.3	ath-miR414	391	CDS	−89	84
		NM_103458.3	ath-miR414	385	CDS	−87	82
		NM_103458.3	ath-miR414	394	CDS	−87	82
		NM_103458.3	ath-miR414	397	CDS	−87	82
	*bZIP68*	NM_102948.4	ath-miR854a-d	7	5′UTR	−96	83
MYB Family	*MYB15*	NM_001035670.1	ath-miR828	683	CDS	−87	80
		NM_001035670.1	ath-miR828	605	CDS	−87	80
	*MYB60*	NM_001331790.1	ath-miR828	582	CDS	−91	84
		NM_001331790.1	ath-miR414	965	CDS	−87	82
		NM_001331790.1	ath-miR414	959	CDS	−87	82
		NM_001331790.1	ath-miR414	971	CDS	−85	80
	*MYB96*	NM_125641.4	ath-miR828	605	CDS	−91	84
		NM_125641.4	ath-miR828	319	CDS	−91	84
	*MYB102*	NM_118264.3	ath-miR828	439	CDS	−96	88
		NM_118264.3	ath-miR829-5p	978	CDS	−85	83
	*MYB108*	NM_111525.4	ath-miR172b,e-5p	1382	CDS	−87	84
		NM_111525.4	ath-miR834	1011	CDS	−91	81
		NM_111525.4	ath-miR858a	837	CDS	−87	80
	*MYB116*	NM_001036014.2	ath-miR858a	293	CDS	−89	82
		NM_001036014.2	ath-miR858a	498	CDS	−89	82
WRKY Family	*WRKY18*	NM_001342115.1	ath-miR781a	1105	3′UTR	−85	83
	*WRKY25*	NM_128578.4	ath-miR403-5p	658	CDS	−87	82
	*WRKY33*	NM_129404.4	ath-miR845a	1167	CDS	−89	81
	*WRKY40*	NM_106732.4	ath-miR838	617	CDS	−87	84
		NM_106732.4	ath-miR156g	282	CDS	−89	82
		NM_106732.4	ath-miR868-5p	437	CDS	−87	82
	*WRKY57*	NM_001334406.1	ath-miR472-5p	328	CDS	−89	82
	*WRKY63*	NM_105331.4	ath-miR855	224	CDS	−89	81
	*WRKY75*	NM_121311.5	ath-miR847	255	CDS	−89	82
		NM_121311.5	ath-miR407	337	CDS	−79	80
HD-ZIP Family	*ATHB7*	NM_001036473.1	ath-miR865-3p	786	CDS	−83	83
	*ATHB54*	NM_001198174.2	ath-miR447a.2-3p	224	5′UTR	−83	83
C2H2 Family	*ZAT10*	NM_102538.3	ath-miR778	623	CDS	−91	83
CAMTA Family	*CAMTA3*	NM_001124889.2	ath-miR844-3p	2623	CDS	−83	81
		NM_001124889.2	ath-miR2112-3p	1938	CDS	−87	80
	*CAMTA6*	NM_001338268.1	ath-miR857	2161	CDS	−83	81
NF-YA Family	*NF-YA1*	NM_001343248.1	miR169 family	1147/1148	3′UTR	−98/−102	85/87
	*NF-YA2*	NM_111443.4	miR169 family	1312/1313	3′UTR	−96/−100	82–89
	*NF-YA3*	NM_001036195.5	miR169 family	1413	3′UTR	−93/−98	80–87
	*NF-YA5*	NM_104294.3	miR169 family	1284	3′UTR	−96/−100	85/87
	*NF-YA6*	NM_001338098.1	miR169 family	1271	3′UTR	−93/−100	80–89
	*NF-YA8*	NM_001198095.1	miR169 family	1505	3′UTR	−98	87
	*NF-YA9*	NM_112983.5	miR169 family	1062/1063	3′UTR	−93/−96	80/83
	*NF-YA10*	NM_001342883.1	miR169 family	735/736	3′UTR	−96/−100	82/89

## Data Availability

The data supporting this study are available from the corresponding author upon reasonable request.

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
