# Peer review of "In Silico Analysis of miRNA-mRNA Binding Sites in *Arabidopsis thaliana* as a Model for Drought-Tolerant Plants"

_plants, 2025, doi:10.3390/plants14121800_

Round 1

Reviewer 1 Report

Comments and Suggestions for Authors

The manuscript by Zhakypbek et al. presents the results of a computational prediction of miRNA-mRNA interactions in the model plant Arabidopsis thaliana. The study examined 274 miRNAs and 48,143 mRNAs using the MirTarget program. The authors identified eight miRNAs targeting seven ROS detoxification genes, 19 miRNAs regulating 13 hormonal signaling genes, and 33 miRNAs targeting transcription factor genes across families including AP2/ERF, NAC, bZIP, MYB, WRKY, HD-ZIP, CAMTA, and NF-YA—all of which are key components of drought-responsive regulatory pathways. The authors present their findings as a foundation for future searches for genes responsible for drought resistance in plants of the Caspian Basin. These results may be of some interest as an indication of potential experimental validation pathways for miRNA regulatory candidates involved in drought resistance.

However, the manuscript has several shortcomings.

  1. Since the authors did not investigate drought-resistant plants of the Caspian region (Halocnemum strobilaceumClimacoptera brachiata, and Artemisia pauciflora), I strongly recommend removing "of the Caspian Region" from the article’s title. In this revised form, the title would appeal to a broader readership. Could the obtained data not also be applied to the study of drought-resistant plants in other regions of the world? Definitely can be. On the other hand, Arabidopsis thaliana and the three aforementioned plant species belong to long-diverged families of flowering plants. This raises serious doubts as to whether the nucleotide sequences of their miRNAs and mRNAs are sufficiently similar to allow data obtained for  thaliana to be extrapolated to H. strobilaceumC. brachiata, and A. pauciflora.
  2. Figure 3 presents the sequence of ath-miR854a-d. This name refers to four closely related but distinct RNAs (a, b, c, d). The figure or text should specify which variant of ath-miR854 is shown. The same applies to ath-miR164a-c in Figure 4A. Figures 3 and 4 also include miRNAs with two variants; in these cases, the sequence of the depicted RNA variant should also be indicated.
  3. The Methods section, which primarily describes the working principle of the MirTarget program, does not justify its selection or compare it with other similar tools. No specific references to the original dataset are provided. The software parameters enabling reproducibility of the results are missing. This section should be rewritten to include all necessary information.

Minor remarks:

Line 44: Scientific plant names should be given in full, without abbreviations.

Line 72: The authors expand the abbreviation "miRNAs" as "mRNA-inhibitory RNAs." While I do not object to this interpretation, the widely accepted expansion—"microRNAs"—should also be mentioned.

Line 159: "Noncanonical mismatches" should be changed to "noncanonical matches."

Author Response

Comments 1: Since the authors did not investigate drought-resistant plants of the Caspian region (Halocnemum strobilaceumClimacoptera brachiata, and Artemisia pauciflora), I strongly recommend removing "of the Caspian Region" from the article’s title. In this revised form, the title would appeal to a broader readership. Could the obtained data not also be applied to the study of drought-resistant plants in other regions of the world? Definitely can be. On the other hand, Arabidopsis thaliana and the three aforementioned plant species belong to long-diverged families of flowering plants. This raises serious doubts as to whether the nucleotide sequences of their miRNAs and mRNAs are sufficiently similar to allow data obtained for  thaliana to be extrapolated to H. strobilaceumC. brachiata, and A. pauciflora.

Response 1: Thank you for the valuable comment. Since we did not analyze Caspian-region plants directly, we have revised the title to: "In Silico Analysis of miRNA–mRNA Binding Sites in Arabidopsis thaliana as a Model for Drought-Tolerant Plants" (Lines: 2–3). This makes the title more accurate and relevant to a broader audience.

Comments 2: Figure 3 presents the sequence of ath-miR854a-d. This name refers to four closely related but distinct RNAs (a, b, c, d). The figure or text should specify which variant of ath-miR854 is shown. The same applies to ath-miR164a-c in Figure 4A. Figures 3 and 4 also include miRNAs with two variants; in these cases, the sequence of the depicted RNA variant should also be indicated.

Response 2: The manuscript has been revised to specify the individual miRNA variants (e.g., ath-miR854a–d, ath-miR164a,b) mentioned in Figures 3 and 4. Their exact binding positions and corresponding free energy values are now clearly indicated in the main text (Lines: 219-224, 230-232, 265, 314-316). Appropriate corrections have been made accordingly. The binding regions and microRNA sequences of these variants are identical (Example figures are included in the attached response file). 

Comments 3: The Methods section, which primarily describes the working principle of the MirTarget program, does not justify its selection or compare it with other similar tools. No specific references to the original dataset are provided. The software parameters enabling reproducibility of the results are missing. This section should be rewritten to include all necessary information.

Response 3: We sincerely thank the reviewer for their valuable comment. To address this, we provide below a comparative table outlining the key features of the MirTarget program in relation to other widely used tools such as psRNATarget, TargetFinder, and RNAhybrid (Example tables are provided in the attached response file). 

Additionally, as examples, the table presents the predicted interactions between the miR-22-5p–ATXN10 and ID00436.3p-miR–ADCYAP1R1 pairs. The binding sites were identified exclusively by the MirTarget program and were not detected by other tools. The MirTarget predictions are located within the CDS and 3′UTR regions of the target mRNAs and demonstrate high thermodynamic stability (ΔG –93 and –104 kJ/mol). The binding schemes also include non-canonical pairings, such as G–U and A–C, which may indicate potential biological relevance (The attached response file includes example tables). 

We appreciate the reviewer’s interest. The comparative table was provided here in the response rather than in the manuscript in order to maintain the overall flow and focus of the Materials and Methods section, which is already densely populated with core experimental details. Since this table is intended as supportive background information rather than a central result, we opted to include it in the response to address the reviewer’s concern directly without overextending the main text.

Regarding the original dataset, all mature microRNA sequences specific to Arabidopsis thaliana were obtained from the miRBase database (v22.1; http://www.mirbase.org), while the corresponding mRNA sequences were retrieved from the NCBI RefSeq database (RefSeq: GCF_000001735.4, (June 2018)) (Lines: 481–483).

In addition, the parameters applied during the use of MirTarget are clearly described in the manuscript. The algorithm's operational principles and parameter configurations are supported by previously published studies, which are also listed [46], [49], [50], [51], [52], [53], [54] (Lines: 494-508).

Minor remarks:

Line 44: Scientific plant names should be given in full, without abbreviations.

Response: The full scientific names (Arabidopsis thaliana, Halocnemum strobilaceum, Climacoptera brachiata, Artemisia pauciflora) were provided at their first mention in both the abstract and the main text. Subsequently, abbreviated forms (e.g., A. thaliana, H. strobilaceum) were used throughout the manuscript to avoid repetition and enhance readability (Lines: 69, 384, 403, 441-442, 456).   

Line 72: The authors expand the abbreviation "miRNAs" as "mRNA-inhibitory RNAs." While I do not object to this interpretation, the widely accepted expansion—"microRNAs"—should also be mentioned.

Response: The abbreviation “miRNAs” has been updated to include the standard term “microRNAs” in the revised text. The phrase now reads: (microRNAs, also referred to as mRNA-inhibitory RNAs), ensuring clarity and alignment with widely accepted terminology (Lines: 73–74).   

Line 159: "Noncanonical mismatches" should be changed to "noncanonical matches."

Response: Thank you for pointing this out. The phrase "noncanonical mismatches" has been corrected to "noncanonical matches" in the revised manuscript to accurately reflect the intended meaning (Lines: 160).   

Reviewer 2 Report

Comments and Suggestions for Authors

The problem of gene expression regulation under drought stress is important for understanding the mechanisms of plant defense against such environmental impacts. The authors conducted a large-scale in silico analysis of miRNA and mRNA interactions. However, several questions arise.

  1. The authors clearly formulate the purpose of the study as an in silico analysis of miRNA and mRNA interactions for drought resistance genes in Arabidopsis thaliana. They further explain that these data can facilitate a comparative analysis of similar mechanisms in other plants. But this may apply to any plants at all. It is unclear why the authors focus only on plants from the Caspian region. Moreover, this clarification is included in the title of the article, which is not justified in the context of the results. The entire work is devoted exclusively to Arabidopsis thaliana, and the application of the obtained results can be stated a priori to any other plants, but not to specific species. Repeated emphasis on plants from the Caspian region in this case does not seem appropriate.
  2. In the Discussion, the authors cite three plants: Halocnemum strobilaceum, Climacoptera brachiate - halophytes and Artemisia pauciflora - xerophyte. All these plants are taxonomically distant from Arabidopsis thaliana, evolved in different environmental conditions and are specifically adapted to a different ecological niche. Their defense mechanisms may differ significantly from the identified variants of miRNA-mRNA interaction, which the authors themselves mention in the last paragraph of the Discussion.
  3. In Materials and Methods, the authors indicate that they used 48,143 A. thaliana genes and 462,274 miRNAs for the study. At the same time, a detailed analysis is provided for "8 miRNAs targeting seven ROS detoxification genes, 19 miRNAs regulating 13 hormonal signaling genes, and 33 miRNAs targeting transcription factor genes". It does not follow from the text of the manuscript what determined the choice of these groups of genes and miRNAs.
  4. The Discussion should describe the mechanisms of miRNA-mRNA binding and the results of such interaction on gene expression, including the influence of the localization of interaction sites on gene functionality.

Author Response

Comments 1: The authors clearly formulate the purpose of the study as an in silico analysis of miRNA and mRNA interactions for drought resistance genes in Arabidopsis thaliana. They further explain that these data can facilitate a comparative analysis of similar mechanisms in other plants. But this may apply to any plants at all. It is unclear why the authors focus only on plants from the Caspian region. Moreover, this clarification is included in the title of the article, which is not justified in the context of the results. The entire work is devoted exclusively to Arabidopsis thaliana, and the application of the obtained results can be stated a priori to any other plants, but not to specific species. Repeated emphasis on plants from the Caspian region in this case does not seem appropriate.

Response 1: We thank the reviewer for the valuable comment. The reference to the Caspian region has been removed from the title, as it does not accurately reflect the scope of the study. The revised title is: "In Silico Analysis of miRNA–mRNA Binding Sites in Arabidopsis thaliana as a Model for Drought-Tolerant Plants" (Lines: 2–3).

The in silico data we obtained can indeed be used for comparative analysis with any other drought-tolerant plant species. In our study, Halocnemum strobilaceum and Climacoptera brachiata (halophytes), as well as Artemisia pauciflora (a xerophyte), were mentioned solely as examples of potential candidates for future investigation due to their natural resilience to drought and salt stress. In response to the reviewer’s comment, we have revised the Discussion section and removed all references specifically mentioning the Caspian region (Lines: 368–479).

Comments 2: In the Discussion, the authors cite three plants: Halocnemum strobilaceum, Climacoptera brachiate - halophytes and Artemisia pauciflora - xerophyte. All these plants are taxonomically distant from Arabidopsis thaliana, evolved in different environmental conditions and are specifically adapted to a different ecological niche. Their defense mechanisms may differ significantly from the identified variants of miRNA-mRNA interaction, which the authors themselves mention in the last paragraph of the Discussion.

Response 2: We sincerely thank the reviewer for this important and well-founded comment. We fully acknowledge that the plants mentioned in the Discussion — Halocnemum strobilaceum, Climacoptera brachiata, and Artemisia pauciflora — are taxonomically distant from Arabidopsis thaliana. Their defense mechanisms may indeed differ due to ecological adaptation to distinct environments. Therefore, in the revised Discussion section, we clarified that applying the results of A. thaliana to these species is presented as a preliminary hypothesis only, intended to guide future comparative studies. We have also emphasized that further conclusions would require functional and experimental validation (Lines: 475–479).

Comments 3: In Materials and Methods, the authors indicate that they used 48,143 A. thaliana genes and 462,274 miRNAs for the study. At the same time, a detailed analysis is provided for "8 miRNAs targeting seven ROS detoxification genes, 19 miRNAs regulating 13 hormonal signaling genes, and 33 miRNAs targeting transcription factor genes". It does not follow from the text of the manuscript what determined the choice of these groups of genes and miRNAs.

Response 3: We thank the reviewer for the insightful comment. While all available A. thaliana genes and miRNAs were initially screened, we focused our detailed analysis on three key functional gene groups: ROS detoxification genes, hormonal signaling genes, and transcription factor genes. These categories were chosen based on their well-documented roles in drought stress responses, as supported by numerous prior studies. The selection was thus hypothesis-driven, aiming to highlight regulatory modules most relevant to abiotic stress adaptation.

Comments 4: The Discussion should describe the mechanisms of miRNA-mRNA binding and the results of such interaction on gene expression, including the influence of the localization of interaction sites on gene functionality.

Response 4: We thank the reviewer for the valuable comment. In response, we have expanded the Discussion section to describe the molecular mechanisms of miRNA–mRNA interactions and their effects on gene expression. These additions provide deeper insights into the functional consequences of miRNA binding site localization within target mRNAs (Lines: 431–438).

Reviewer 3 Report

Comments and Suggestions for Authors

Review on the manuscript “ In Silico Analysis of miRNA-mRNA Binding Sites in Arabidopsis thaliana as a Model for Drought-Tolerant Plants of the Caspian Region”

 The study and identification of molecular mechanisms of plant tolerance to drought, has important scientific and practical, especially in arid and semi-arid areas.

The manuscript presents the results of bioinformatics analysis (in silico) of miRNA–mRNA interactions regulating stress-responsive genes in A. thaliana, which provide information on possible post-transcriptional regulatory networks in antioxidant defense, hormonal signaling and transcription factor pathways.

The authors identified eight miRNAs targeting seven ROS detoxification genes, 19 miRNAs regulating 13 hormone signaling genes, and 33 miRNAs targeting transcription factor genes in stress-responsive A. thaliana families.

Unfortunately, the manuscript does not present any results for wild species of the Caspian region H. strobilaceum, C. brachiata and A. pauciflor. The authors' attempts to interpolate the results obtained for A. thaliana to these species are questionable and speculative.

Comments

Abstract

  1. It is not clear from the abstract what Arabidopsis thaliana has to do with the Caspian region
  2. The sentence «These findings suggest conserved miRNA-mediated mechanisms across drought-tolerant species of the Caspian region, including H. strobilaceum, C.brachiata, and A. pauciflora» does not provide insight into the relationship of the results obtained on thaliana with species of H. strobilaceum, C.brachiata, and A. pauciflora
  3. It is not clear from the abstract which results were obtained for strobilaceum, C. brachiata, and A. pauciflor
  4. The name of the species must be complete when first used.

Results

  1. In Tables 1 and 2, the name of the protein/function that the genes encode should be indicated (or in a footnote to the tables).

Discussion

  1. 365 – “The present study provides an extensive in silico analysis of miRNA–mRNA inter- actions in A. thaliana as a model for drought-tolerant plants of the Caspian region.” – incorrect sentence, since A. thaliana is a model for any plant’s investigations, due to its best study, and not specifically for drought-tolerant plants of the Caspian region. Moreover, A. thaliana is not a drought-tolerant species.
  2. It is unclear from the text provided, why these three species, information about which appears only in the discussion section (lines 377, 391, 419), were selected from the entire flora of the Caspian region. These species have fundamentally different strategies for adaptation to osmotic stress. Halocnemum strobilaceum is a succulent salt-accumulating hyperhalophyte, brachiate is a xerohalophyte with C4 type of photosynthesis, A. pauciflora is a hyperxerophyte with very specific mechanisms of drought resistance.
  3. 377-380.  “The arid and saline ecosystems of the Caspian region are home to highly stress-resilient species like Halocnemum strobilaceum (H. strobilaceum), a halophyte capable of surviving in soils with salinity up to 38% and enduring prolonged drought. Its exceptional resilience implies the presence of robust cellular defense mechanisms, including osmotic adjustment and ROS detoxification [33 ]”  -- The title of reference 33 does not mention the species H. strobilaceum , but the genus Climacoptera. And I could not find this article on the internet.
  4. 384. “Given the conserved miR398 – CSD1 regulatory module observed in A. thaliana, it is plausible that similar miRNA-mediated regulation operates in H. strobilaceum.” –-- This is a speculative statement.
  5. 395-398. “Our A. thaliana based in silico predictions … ath-miR167c-5p regulate ARF6, a pivotal auxin response factor. If a functional homolog of ARF6 is present in C. brachiata, its regulation by miR167 family miRNAs could represent a conserved drought-adaptive mechanism across species.” –- This is a speculative statement.
  6. 427 “It is plausible that A. pauciflora conserves analogous miRNA–transcript interaction mechanisms—such as miR169 family members targeting NF-YA genes that modulate gene expression under drought conditions.” –- This is a speculative statement.
  7. 453. “This study elucidates the regulatory network underlying miRNA-mediated drought tolerance in A. thaliana, uncovering conserved modules potentially shared with drought and salt-tolerant species of the Caspian region.” –-This is a controversial statement.

First, the methods do not indicate that the authors studied the genes of A. thaliana as part of a drought stress experiment, i.e. that the A. thaliana plants were exposed to drought and therefore a “drought response” was implemented.

Secondly, A. thaliana is not a drought-resistant and salt-tolerant species, unlike the flora of the Caspian region.

If the authors believe that they have uncovered “conservative modules” (for ROS detoxification, hormonal signaling, drought-responsive transcriptional control) that are inherent in the mesophyte A. thaliana, then why is it that for H. strobilaceum, C. brachiata, and A. pauciflora, with different strategies for adaptation to drought, they assume the presence of only one of the three: for H. strobilaceum - only ROS detoxification, C. brachiata - only hormonal signaling, A. pauciflora – only transcriptional control.

Theoretically, all identified "conservative modules" could be present in these species. But this needs to be proven.

In my opinion, the manuscript in its present form cannot be recommended for publication.

Author Response

Abstract

Comments 1: It is not clear from the abstract what Arabidopsis thaliana has to do with the Caspian region   

Response 1: Thank you for your insightful comment. The abstract has been revised to clarify the rationale for using Arabidopsis thaliana as a model organism (Lines: 28–30). Due to its fully sequenced genome and extensive molecular characterization, Arabidopsis thaliana serves as a well-established reference species for studying gene regulatory networks under abiotic stress conditions, including drought and salinity. These attributes make it particularly suitable for generating in silico predictions that can inform comparative analyses in non-model, stress-adapted plant species.

Additionally, the title and main text have been revised to remove region-specific references such as the "Caspian region." We have revised the title to: "In Silico Analysis of miRNA–mRNA Binding Sites in Arabidopsis thaliana as a Model for Drought-Tolerant Plants" (Lines: 2–3). This makes the title more accurate and relevant to a broader audience.

Comments 2: The sentence «These findings suggest conserved miRNA-mediated mechanisms across drought-tolerant species of the Caspian region, including H. strobilaceum, C.brachiata, and A. pauciflora» does not provide insight into the relationship of the results obtained on thaliana with species of H. strobilaceum, C.brachiata, and A. pauciflora.

Response 2: We thank the reviewer for the helpful comment. The species H. strobilaceum, C. brachiata, and A. pauciflora were discussed in the manuscript not as direct subjects of analysis, but as representative examples of drought-adapted plants. These species were included in the Discussion section to illustrate the potential relevance of the findings, as genomic and transcriptomic resources for such non-model plants from arid and saline ecosystems remain limited. Therefore, Arabidopsis thaliana was used as a well-established model organism to generate in silico predictions, which serve as a hypothesis-building framework. The aim was to highlight how the predicted regulatory modules in A. thaliana could offer insights for future experimental validation and comparative studies in ecologically important but genetically undercharacterized species (Lines: 40–43).

Comments 3: It is not clear from the abstract which results were obtained for strobilaceum, C. brachiata, and A. pauciflor

Response 3: As correctly noted, the study is based on Arabidopsis thaliana as a model species, and no direct experimental results were obtained for Halocnemum strobilaceum, Climacoptera brachiata, or Artemisia pauciflora. To clarify this, we have now revised the abstract to explicitly state that these drought-tolerant species were included only as examples of ecologically relevant plants for future comparative research. Their mention is intended to frame the potential translational relevance of the conserved miRNA–mRNA interactions observed in A. thaliana, not to suggest that original data were generated for them.

These findings suggest conserved miRNA-mediated mechanisms that may also be relevant to drought-adapted species such as halophytes and xerophytes, including Halocnemum strobilaceum, Climacoptera brachiata, and Artemisia pauciflora. While no direct experimental data were obtained for these species, the results from Arabidopsis thaliana offer a comparative genomic framework for hypothesis generation and future functional studies (Lines: 28–45).

Comments 4: The name of the species must be complete when first used.

Response 4: The full scientific names (Arabidopsis thaliana, Halocnemum strobilaceum, Climacoptera brachiata, Artemisia pauciflora) were provided at their first mention in both the abstract and the main text. Subsequently, abbreviated forms (e.g., A. thaliana, H. strobilaceum) were used throughout the manuscript to avoid repetition and enhance readability (Lines: 69, 384, 403, 441-442, 456).  We thank the reviewer for helping us improve the clarity and scientific rigor of the manuscript.

Results

Comments 5: In Tables 1 and 2, the name of the protein/function that the genes encode should be indicated (or in a footnote to the tables).

Response 5: We have updated Tables 1 and 2 by adding a column titled "Protein / Function", which specifies the protein names or biological functions encoded by the listed genes (Lines: 128–130, 208-209).

Discussion

Comments 6: 365 – “The present study provides an extensive in silico analysis of miRNA–mRNA inter- actions in A. thaliana as a model for drought-tolerant plants of the Caspian region.” – incorrect sentence, since A. thaliana is a model for any plant’s investigations, due to its best study, and not specifically for drought-tolerant plants of the Caspian region. Moreover, A. thaliana is not a drought-tolerant species.

Response 6: We agree that Arabidopsis thaliana is not inherently a drought-tolerant species and is widely recognized as a general model organism in plant molecular biology. To address this, we have revised the sentence in the Discussion section (Lines: 368–370).

Comments 7: It is unclear from the text provided, why these three species, information about which appears only in the discussion section (lines 377, 391, 419), were selected from the entire flora of the Caspian region. These species have fundamentally different strategies for adaptation to osmotic stress. Halocnemum strobilaceum is a succulent salt-accumulating hyperhalophyte, brachiate is a xerohalophyte with C4 type of photosynthesis, A. pauciflora is a hyperxerophyte with very specific mechanisms of drought resistance.

Response 7: We sincerely thank the reviewer for this important remark. We acknowledge that Halocnemum strobilaceum, Climacoptera brachiata, and Artemisia pauciflora are not model organisms and that each of them possesses fundamentally different mechanisms of adaptation to osmotic stress. These species were not the subject of direct analysis in this study but were mentioned in the Discussion section solely as ecologically relevant examples of drought- and salt-tolerant plants. Their inclusion was intended to illustrate the potential applicability of the predicted miRNA–mRNA regulatory modules identified in Arabidopsis thaliana to other species exhibiting distinct stress adaptation strategies (Lines: 384–389, 403-412, 441-452).

Accordingly, the wording in the manuscript has been revised: the phrase model for drought-tolerant plants of the Caspian region has been changed to model for drought-tolerant plants.

Comments 8: 377-380.  “The arid and saline ecosystems of the Caspian region are home to highly stress-resilient species like Halocnemum strobilaceum (H. strobilaceum), a halophyte capable of surviving in soils with salinity up to 38% and enduring prolonged drought. Its exceptional resilience implies the presence of robust cellular defense mechanisms, including osmotic adjustment and ROS detoxification [33 ]”  -- The title of reference 33 does not mention the species H. strobilaceum , but the genus Climacoptera. And I could not find this article on the internet.

Response 8: The sentence in question has been revised and supported with updated scientific references. The new version highlights the salinity and drought tolerance of Halocnemum strobilaceum, based on the following peer-reviewed sources:

- [33. Qu, X.X. et al., Ann. Bot. 2008, 101(2), 293–299], which reports that H. strobilaceum can naturally grow in soils containing up to 38% total salt (dry weight);

- [34. Barbafieri, M. et al., Plants (Basel), 2023, 12(9), 1737], which describes the species’ physiological adaptation to high salinity.

(Lines: 384–389).

Comments 9: 384. “Given the conserved miR398 – CSD1 regulatory module observed in A. thaliana, it is plausible that similar miRNA-mediated regulation operates in H. strobilaceum.” –-- This is a speculative statement.

Response 9: Thank you for your comment. We agree that the original sentence may appear speculative. To address this, we have revised the statement to adopt a more cautious and scientifically appropriate tone. This change reflects the hypothetical nature of the statement while maintaining its relevance to the discussion on potential conservation of stress-responsive modules (Lines: 391–400).

Comments 10: 395-398. “Our A. thaliana based in silico predictions … ath-miR167c-5p regulate ARF6, a pivotal auxin response factor. If a functional homolog of ARF6 is present in C. brachiata, its regulation by miR167 family miRNAs could represent a conserved drought-adaptive mechanism across species.” –- This is a speculative statement.

Response 10: Considering the speculative nature of the original sentence, we have revised the paragraph accordingly. The updated version explicitly states that no transcriptomic or genomic data are currently available for C. brachiata, and the proposed assumption is now framed as a scientific hypothesis. Additionally, we emphasized that this hypothesis requires future experimental validation in non-model halophytic plants. We hope that this revision improves the scientific clarity of the manuscript while maintaining the relevance of the proposed regulatory module as a potential direction for future research (Lines: 406–412).

Comments 11: 427 “It is plausible that A. pauciflora conserves analogous miRNA–transcript interaction mechanisms—such as miR169 family members targeting NF-YA genes that modulate gene expression under drought conditions.” –- This is a speculative statement.

Response 11: To address the speculative nature of the original sentence, we have revised it to reflect a more cautious and hypothesis-based formulation. The updated version explicitly states the lack of transcriptomic data for A. pauciflora and frames the assumption regarding the conservation of miR169 - NF-YA interactions as a hypothesis. Additionally, we emphasize the need for future experimental validation. This adjustment improves the scientific rigor of the statement while retaining its relevance to the discussion of potential drought-adaptive regulatory modules (Lines: 449–452).

Comments 12: 453. “This study elucidates the regulatory network underlying miRNA-mediated drought tolerance in A. thaliana, uncovering conserved modules potentially shared with drought and salt-tolerant species of the Caspian region.” – This is a controversial statement.

Response 12: Thank you for your comment. To address the concern regarding the potentially controversial nature of the original sentence, we have revised it to adopt a more cautious, hypothesis-based tone. The updated version avoids specifying geographic regions and presents the findings as a potential framework for understanding conserved stress-responsive mechanisms, with an emphasis on the need for future experimental validation (Lines: 475–479).

Comments: First, the methods do not indicate that the authors studied the genes of A. thaliana as part of a drought stress experiment, i.e. that the A. thaliana plants were exposed to drought and therefore a “drought response” was implemented.

Response: As noted, the current study does not involve direct drought treatment of A. thaliana plants. Instead, it is based on publicly available genomic resources and in silico bioinformatics analysis to predict miRNA–mRNA interactions. The selected genes are those previously reported to be involved in drought response, and the predicted miRNAs were analyzed accordingly. Therefore, the drought response was not experimentally induced but rather inferred through computational prediction.

Comments: Secondly, A. thaliana is not a drought-resistant and salt-tolerant species, unlike the flora of the Caspian region.

Response: Thank you for your comment. We fully acknowledge that A. thaliana is not a naturally drought- or salt-tolerant species, unlike the flora of the Caspian region. However, A. thaliana is widely used as a model organism due to its well-annotated genome and extensive molecular data resources. In this study, it was utilized for in silico prediction of stress-responsive regulatory modules. The results are not intended to directly compare A. thaliana with halophytic or xerophytic species, but rather to explore the potential conservation of such regulatory mechanisms in stress-adapted plants. This point has now been clarified in the manuscript.

Comments: If the authors believe that they have uncovered “conservative modules” (for ROS detoxification, hormonal signaling, drought-responsive transcriptional control) that are inherent in the mesophyte A. thaliana, then why is it that for H. strobilaceum, C. brachiata, and A. pauciflora, with different strategies for adaptation to drought, they assume the presence of only one of the three: for H. strobilaceum - only ROS detoxification, C. brachiata - only hormonal signaling, A. pauciflora – only transcriptional control.

Response: Thank you for this thoughtful observation. While A. thaliana served as a model in which multiple drought-related regulatory modules were identified (ROS detoxification, hormonal signaling, and transcriptional control), our predictions for H. strobilaceum, C. brachiata, and A. pauciflora were limited by the availability of gene and miRNA data specific to each species.

Our intention was not to suggest that these species possess only one such mechanism, but rather to highlight specific miRNA–mRNA modules that could represent conserved drought-related regulatory strategies, based on the available data. We agree that broader datasets may reveal the coexistence of multiple modules, and we have clarified this limitation in the revised text.

Comments: Theoretically, all identified "conservative modules" could be present in these species. But this needs to be proven.

Response: For each species, only specific modules were highlighted due to limitations in available data. Theoretically, all identified conserved modules could be present in these species, but this hypothesis requires experimental validation. This point has been clarified in the revised manuscript. Thank you for your valuable comment.

Round 2

Reviewer 2 Report

Comments and Suggestions for Authors

The authors took into account all the recommendations and made corrections to the text of the manuscript. I hope that these changes have improved the quality of the manuscript and will allow readers to better understand the presented research results.

Author Response

Comments: The authors took into account all the recommendations and made corrections to the text of the manuscript. I hope that these changes have improved the quality of the manuscript and will allow readers to better understand the presented research results.

Response: Thank you for your positive feedback. We appreciate your time and are glad the revisions have improved the manuscript's clarity and quality

Reviewer 3 Report

Comments and Suggestions for Authors

Review on the revised manuscript “In Silico Analysis of miRNA-mRNA Binding Sites in Arabidopsis thaliana as a Model for Drought-Tolerant Plants”

The authors have revised the text using more cautious formulation regarding species Halocnemum strobilaceumClimacoptera brachiata, and Artemisia pauciflora.

- However, in the text of the manuscript it remains unclear why these three species were chosen.

(Comments 7: It is unclear from the text provided, why these three species, information about which appears only in the discussion section (lines 377, 391, 419), were selected from the entire flora of the Caspian region. These species have fundamentally different strategies for adaptation to osmotic stress. Halocnemum strobilaceum is a succulent salt-accumulating hyperhalophyte, brachiate is a xerohalophyte with C4 type of photosynthesis, A. pauciflora is a hyperxerophyte with very specific mechanisms of drought resistance.)

- In addition, the authors are being disingenuous when they refer to articles [33] and [34], which studied the effect of salinity (as well as temperature, and light) on H. strobilaceum, and not drought, which the authors write about on line 386, and in general the entire manuscript is devoted to drought.

(Comments 8: 377-380.  “The arid and saline ecosystems of the Caspian region are home to highly stress-resilient species like Halocnemum strobilaceum (H. strobilaceum), a halophyte capable of surviving in soils with salinity up to 38% and enduring prolonged drought. Its exceptional resilience implies the presence of robust cellular defense mechanisms, including osmotic adjustment and ROS detoxification [33 ]”  -- The title of reference 33 does not mention the species H. strobilaceum , but the genus Climacoptera. And I could not find this article on the internet.

Response 8: The sentence in question has been revised and supported with updated scientific references. The new version highlights the salinity and drought tolerance of Halocnemum strobilaceum, based on the following peer-reviewed sources:

- [33. Qu, X.X. et al., Ann. Bot. 2008, 101(2), 293–299], which reports that H. strobilaceum can naturally grow in soils containing up to 38% total salt (dry weight);

- [34. Barbafieri, M. et al., Plants (Basel), 2023, 12(9), 1737], which describes the species’ physiological adaptation to high salinity. (Lines: 384–389).)

[33] Qu, X.X.; Huang, Z.Y.; Baskin, J.M.; Baskin, C.C. Effect of Temperature, Light and Salinity on Seed Germination and Radicle Growth of the Geographically Widespread Halophyte Shrub Halocnemum strobilaceum. Ann. Bot. 2008, 101(2), 293–299.

[34] Barbafieri M., Bretzel F., Scartazza A., Di Baccio D., Rosellini I., Grifoni M., Pini R., Clementi A., Franchi E. Response to hypersalinity of four halophytes growing in hydroponic floating systems: prospects in the phytomanagement of high saline wastewaters and extreme environments. Plants (Basel). 2023, 12(9), 1737.

- The authors are being disingenuous also in relation to C. brachiata

405-406 – “Its role in soil stabilization and its drought tolerance are likely supported by ABA signaling and the accumulation of protective metabolites [34,36].”  –-- Article [34] shows the effects of hypersaline conditions to some species, but Climacoptera species are not among them.

[34] Barbafieri M., Bretzel F., Scartazza A., Di Baccio D., Rosellini I., Grifoni M., Pini R., Clementi A., Franchi E. Response to hypersalinity of four halophytes growing in hydroponic floating systems: prospects in the phytomanagement of high saline wastewaters and extreme environments. Plants (Basel). 2023, 12(9), 1737.  

406-409 -  “Although no transcriptomic or genomic data confirming the presence of an ARF6 homolog in C. brachiata are currently available, it is plausible that auxin-responsive regulatory modules may also function in this species, given its physiological resilience in arid and saline environments [37].”  –-- Article [37] shows the effect of salinity on Suaeda and Salicornia species, which are characterized by other mechanisms of adaptation to salinity.

[37] Benjamin J.J., Miras-Moreno B., Araniti F., Salehi H., Bernardo L., Parida A., Lucini L. Proteomics revealed distinct responses to salinity between the halophytes Suaeda maritima (L.) Dumort and Salicornia brachiata (Roxb). Plants 2020, 9, 227.  

That is, the authors found articles on the effect of salinity on the species of interest to them or other species of Chenopodiaceae. Although there is an osmotic component in salt stress, and similar defense mechanisms can be realized at a certain time of exposure, but in general, reactions to salinity and drought differ significantly. In addition, the mechanisms of adaptation to salinity and drought in different species can vary greatly (as the authors themselves write), so it is also incorrect to extrapolate the obtained results to different species.

In my opinion, since there is neither relevant literary data nor experiments on wild species, the discussion on them should be greatly reduced.

Author Response

Comment: The authors have revised the text using more cautious formulation regarding species Halocnemum strobilaceum, Climacoptera brachiata, and Artemisia pauciflora.

- However, in the text of the manuscript it remains unclear why these three species were chosen.

(Comments 7It is unclear from the text provided, why these three species, information about which appears only in the discussion section (lines 377, 391, 419), were selected from the entire flora of the Caspian region. These species have fundamentally different strategies for adaptation to osmotic stress. Halocnemum strobilaceum is a succulent salt-accumulating hyperhalophyte, brachiate is a xerohalophyte with C4 type of photosynthesis, A. pauciflora is a hyperxerophyte with very specific mechanisms of drought resistance.).

Response: We thank the reviewer for the valuable comment. The three plant species (Halocnemum strobilaceum, Climacoptera brachiata, and Artemisia pauciflora) were originally included as representative examples of stress-adapted flora when the manuscript had a working title referring to the Caspian region. Following earlier reviewer suggestions, all references to the Caspian region were removed. Consequently, the presence of these species in the revised text may have appeared unclear or unsupported. To ensure clarity and consistency, the mentioned plant species and all related references have now been completely removed from the discussion section (Lines 368–440).

Comment: - In addition, the authors are being disingenuous when they refer to articles [33] and [34], which studied the effect of salinity (as well as temperature, and light) on H. strobilaceum, and not drought, which the authors write about on line 386, and in general the entire manuscript is devoted to drought.

(Comments 8: 377–380.  “The arid and saline ecosystems of the Caspian region are home to highly stress-resilient species like Halocnemum strobilaceum (H. strobilaceum), a halophyte capable of surviving in soils with salinity up to 38% and enduring prolonged drought. Its exceptional resilience implies the presence of robust cellular defense mechanisms, including osmotic adjustment and ROS detoxification [33 ]”  -- The title of reference 33 does not mention the species H. strobilaceum , but the genus Climacoptera. And I could not find this article on the internet.

Response: We cited these references solely to illustrate the species’ general ability to adapt to stress conditions. However, to avoid any misunderstanding, these citations have been removed from the discussion section.

Comment: - The authors are being disingenuous also in relation to C. brachiata

405-406 – “Its role in soil stabilization and its drought tolerance are likely supported by ABA signaling and the accumulation of protective metabolites [34,36].”  –-- Article [34] shows the effects of hypersaline conditions to some species, but Climacoptera species are not among them.

Response: We cited this reference to provide general contextual information about the adaptation of related plant taxa to arid and saline environments. However, to avoid potential misunderstanding, this citation has been removed from the discussion section.

Comment: That is, the authors found articles on the effect of salinity on the species of interest to them or other species of Chenopodiaceae. Although there is an osmotic component in salt stress, and similar defense mechanisms can be realized at a certain time of exposure, but in general, reactions to salinity and drought differ significantly. In addition, the mechanisms of adaptation to salinity and drought in different species can vary greatly (as the authors themselves write), so it is also incorrect to extrapolate the obtained results to different species.

In my opinion, since there is neither relevant literary data nor experiments on wild species, the discussion on them should be greatly reduced.

Response: We thank the reviewer for this valid comment. Considering the lack of specific data on wild species, their descriptions have been removed from the discussion. The results are now limited to hypotheses based solely on Arabidopsis thaliana.

Round 3

Reviewer 3 Report

Comments and Suggestions for Authors

The authors took into account the comments and revised the text.